# Mental health during the COVID-19 pandemic: Impacts of disease, social isolation, and financial stressors

**Robert E. Kraut**[1]*, **Han Li**[2], **Haiyi Zhu**[3]

**1** Human-Computer Interaction Institute & Tepper School of Business, Carnegie Mellon University, Pittsburgh, Pennsylvania, United States of America, **2** School of Journalism, Fudan University, Shanghai, China, **3** Human-Computer Interaction Institute, Carnegie Mellon University, Pittsburgh, Pennsylvania, United States of America

* robert.kraut@cmu.edu

**Data Availability Statement:** Most of the data reported in this paper are publicly available at a Github repository maintained by the Delphi research group at Carnegie Mellon University and available through R and python APIs (https://cmu-

## Abstract

### Background

Although research shows that the Covid-19 pandemic has led to declines in mental health, the existing research has not identified the pathways through which this decline happens.

### Aims

The current study identifies the distinct pathways through which COVID-induced stressors (i.e., social distancing, disease risk, and financial stressors) trigger mental distress and examines the causal impact of these stressors on mental distress.

### Methods

We combined evidence of objective pandemic-related stressors collected at the county level (e.g., lack of social contact, infection rates, and unemployment rates) with self-reported survey data from over 11.5 million adult respondents in the United States collected daily for eight months. We used mediation analysis to examine the extent to which the objective stressors influenced mental health by influencing individual respondents' behavior and fears.

### Results

County-level, day-to-day social distancing predicted significantly greater mental distress, both directly and indirectly through its effects on individual social contacts, worries about getting ill, and concerns about finances. Economic hardships were indirectly linked to increased mental distress by elevating people's concerns about their household's finances. Disease threats were both directly linked to mental distress and indirectly through its effects on individual worries about getting ill. Although one might expect that social distancing from people outside the home would have a greater influence on people who live alone, sub-analyses based on household composition do not support this expectation.

delphi.github.io/delphi-epidata/api/covidcast.html).
These data come from multiple sources, with data
licensing handled separately for each source. The
de-identified survey data are available to
researchers associated with universities or non-
profit organizations. Researchers who want access
to the survey data should submit an information
request on Facebook's COVID-19 Symptom Survey
– Request for Data Access page (https://
dataforgood.facebook.com/dfg/tools/covid-19-
trends-and-impact-survey#accessdata.)

**Funding:** rek NSF BCS 2030074 US National
Science Foundation https://www.nsf.gov/ The
funders had no role in study design, data collection
and analysis, decision to publish, or preparation of
the manuscript.

**Competing interests:** I have read the journal's
policy and the authors of this manuscript have the
following competing interests: Robert Kraut is a
part-time employee of Meta, the company that
partnered with Carnegie Mellon University to
collect the data. Kraut had no role in data collection
and as part of the data-use agreement was not
allowed direct access to the data.

## Conclusion

This research provides evidence consistent with the thesis that the COVID-19 pandemic harmed the mental well-being of adults in the United States and identifies specific stressors associated with the pandemic that are responsible for increasing mental distress.

## Introduction

By July 2022, the COVID-19 pandemic had infected over 88 million people in the United States and caused over a million deaths [1]. The Lancet's COVID-19 Commission on Mental Health Task Force concluded that there was clear evidence that psychological distress increased during the early months of COVID-19 and that the pandemic was harming mental health [2]. Vahratian and colleagues at the National Center for Health Statistics at the CDC, using cross-sectional surveys from almost 800,00 respondents, documented a 14% increase in the number of US adults experiencing symptoms of anxiety or depression in the seven days prior to its surveys [3].

However, the existing research has not convincingly identified the distinct pathways through which COVID-induced stressors trigger mental distress nor disentangled the causal relationships among them. Most research has treated the pandemic as an undifferentiated whole, showing for example that indicators of mental distress were higher in locations [4] or at times with higher infection rates [5–7] or among specific groups of people, such as healthcare professionals [8]. While this type of research demonstrates that the pandemic was associated with increases in mental distress, it rarely differentiates distinct stressors associated with the pandemic and how they influence mental distress.

A small number of studies have attempted to differentiate the influence of distinct stressors associated with the pandemic, such as risks of disease and death, unemployment and loss of income, and social isolation resulting from stay-at-home policies and individual choices, and even fewer have attempted to examine the impact of these stressors simultaneously. We are aware only of recent research by Kämpfen et al [9], who used a large national probability survey of US adults conducted for three weeks in March 2020 to examine the extent to which disease and financial or social stressors predicted changes in symptoms of anxiety and depression. The stressors respectively were respondents' perceived risk of getting infected and dying from Covid-19, their concerns that they would run out of money, and self-reported reductions in their social activities. They found that all three of these stressors predicted worse anxiety and depression outcomes, as measured by higher PHQ-4 scores [10] after controlling for relevant demographic variables, including sex, age, educational, race, and marital status. Alimoradi et al [11] found that sleep problems appear to have been common during the COVID-19 pandemic and were associated with higher levels of psychological distress among the general population, healthcare professionals, and COVID-19 patients.

Although an excellent start, this research has important limitations that undercut the conclusion that these stressors cause increases in anxiety and depression. Perhaps the most important are the related problems of endogeneity, common method bias, and reverse causation. Both the stressors and the mental health outcome were measured via respondents' self-reports. Therefore, it is plausible that people who had higher levels of generalized anxiety and depression also perceived greater risk and had greater anxiety from the specific stressors the survey targeted. That is, respondents' generalized mental distress may have led them to perceive

higher risks independent of their objective risk. Similarly, those with higher levels of generalized anxiety and depression may have been more likely to practice social distancing.

The main substantive goal of our research is to examine the causal impact of distinct pandemic-related stressors on mental health. We examine the disease, financial, and social stressors identified by Kämpfen et al [9]. Our methodological goal is to reduce threats to causal interpretation by combining evidence of **objective pandemic-related stressors** with **self-reported survey data** and **using mediation analysis** to examine the extent to which the objective stressors influence mental health by influencing individual respondents' behavior and fears. In contrast to most previous research, we examine the relationship between changes in pandemic-related stressors and mental health not just at the beginning of the pandemic but over a long, 8-month period and collect population-weighted data from over 11.5 million US adult respondents.

A simplified path model is presented in Fig 1 summarizing our hypotheses about how objective community-level stressors (i.e., social distancing, disease severity, and financial stressors) predicted mental distress, mediated by individual respondents' social isolation and worries about disease and finances. We hypothesize that the pandemic could plausibly harm mental health through three distinct routes. First, the pandemic could increase mental distress by increasing fears of getting the disease [12]. Second, pandemic-induced social distancing, a result of both public health recommendations and individual decisions to limit exposure, could influence mental distress in a more complicated way. On the one hand, the hypothesis that social distancing would increase mental distress is based on decades of research demonstrating that social contact is associated with better physical and mental health and that loneliness and social isolation are associated with increased depression and mortality [13–15]. On the other hand, social distancing could also reduce people's exposure to disease, which in turn could reduce their fears about getting ill and their overall mental distress. In addition, social distancing could also reduce employment opportunities for some people, such as restaurant workers, but increase employment for other occupations, such as warehouse workers or delivery drivers, or lower commuting and other work-related expenses for others, such as white-

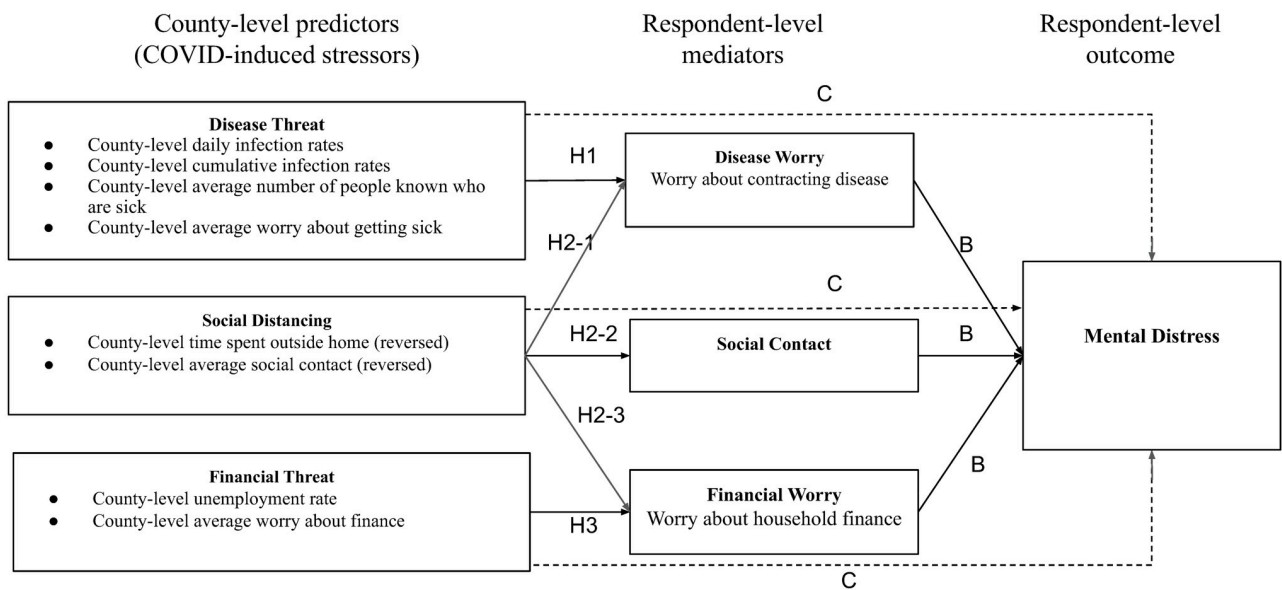

**Fig 1. Predicted relationships among county-level COVID-related stressors, respondent-level mediators, and respondents' mental distress.**

collar workers. Thus, to better understand the impact of social distancing policies and norms, we hypothesize that county-level social distancing can also influence mental distress by influencing personal social contact, illness worries, and financial worries. Third, the pandemic could lead to financial hardships and fears about them caused by economic slowdowns, which in turn could cause mental distress [16].

The solid lines labeled 'A' represent the degree to which county-level distancing and stressors influence respondents' social contact and their individual worries, and the solid lines labeled 'B' represent the association between respondents' social contact and their individual worries, and their mental distress. The dotted lines labeled 'C' represent the direct pathway between the county-level stressors and respondents' mental distress. The product of 'A' and 'B' represents indirect effects of the county-level stressors on distress, which are mediated by their effects on respondents' social contact and their individual worries about the illness and household finances.

## Methods

### Study design and data collection

We used time-series data from the COVID-19 Trends and Impact Survey [17], a collaborative project between Facebook and academics to support COVID-19 research. Each day Facebook invited a random sample of users in the United States at least 18 years old to take a survey designed and collected by Carnegie Mellon University's (CMU) Delphi Group by placing a notification at the top of their News Feed. The survey was anonymous and did not collect any personally identifiable information.

The current paper relies on 11,974,779 survey responses from 239 cross-sectional samples gathered daily from April 6 to November 30, 2020. Although some respondents may have taken the survey more than once, because of the anonymous nature of the data collection we cannot identify surveys completed by the same person, and we therefore treat them as independent. Each survey response was weighted to adjust for non-response and coverage biases so that the distribution of age, gender, and county of residence in the survey samples were representative of the general population of the United States. (For details of the weighting see [17]). The demographic characteristics of the weighted sample closely matches data from the 2019 American Community Survey (ACS), except for an overrepresentation of highly educated respondents (See S1 Table).

The survey asked respondents questions about COVID-19 symptoms they and other household members had. It also asked questions about the depression, anxiety, in-person social contact, and COVID-related stressors respondents were experiencing, the focus of this paper. Respondents also described their demographic characteristics, including their household composition, their gender, and their approximate age.

The survey data included respondents' Federal Information Processing System (FIPS) county code, which allowed responses to be joined with county-level data from multiple sources describing daily and cumulative infection rates of COVID-19 in the county [17], unemployment rates [18], and the time county residents spent outside of the household as estimated from mobile devices [19]. We also estimated from the survey data daily county-level worries about illness and finances and county-level social contact by first removing the respondent's data and then calculating the 2-week moving average surrounding the respondent's survey date. These county-level variables are treated as potential causes that could influence mental distress directly or indirectly, by influencing the respondent-level predictors of interest, including social contact and worries about disease and finances.

### Ethics statement

The Carnegie Mellon University Institutional Review Board (IRB) reviewed the research plan and granted approval under exempt review on 7/10/2020, (study id STUDY2020_00000292).

### Measures

Table 1 below describes the measures of all variables used in the analyses.

### Analysis logic

We used structural equation modeling, done with the Stata SEM package [20], to implement the mediation analysis represented in Fig 1's path diagram. The maximum likelihood estimation was used in the analysis. The mediation analysis examines the potential causal pathways through which day-to-day variation in objective, county-level disease threat, financial threat, and social distancing could influence respondents' mental distress either directly or mediated by respondents' individual social contact and worries about disease and finances. To control for static characteristics of respondents' locations, we first centered each variable by the county-identification FIPS code, which removes all fixed effects associated with that geographic area, such as its size, economic prosperity, demographics, and other unmeasured static differences. What remains is the day-to-day variability in respondents' mental distress, disease and financial worries, and county-level stressors.

The use of both respondent-level variables collected via the survey and county-level ones from multiple sources reduces many threats to assessing causation from observational data. Because the variation in county-level stressors and social distancing are outside of any single respondent's control, we can treat them as exogenous variables that can directly influence respondent-level variables but not be caused by them, eliminating possibilities of reverse causation. For example, changes in county-level disease threat (e.g., infection rates) and financial threat (e.g., unemployment rates) are exogenous variables that could influence respondents' mental distress directly or indirectly through their worries about getting ill or their concerns about household finances. Similarly, changes in the time residents of a county spent outside their home, reflecting both government regulations and evolving local norms, are exogenous variables that could influence a respondent's mental distress both directly or indirectly through its effect on their social distancing behavior and their worries about disease and finances. The multiple levels of analysis reduce the likelihood that unobservable variables, like respondents' socio-economic status, work-status, or disability, jointly influence their social contact and their mental distress. The multiple levels of analysis also reduce common-method bias, in which associations are inflated because potential causes (e.g., financial worries) and consequence (e.g., mental distress) are measured using similar measures from a single source [21].

## Results

Tables 2 and 3 below show descriptive statistics and correlations among the variables.

Fig 2 shows how self-reported mental distress (depression and anxiety), county-level predictors and mediators varied over time. In addition to these time-varying variables, analyses reported below included several static covariates: respondents' self-reported gender, approximate age, and whether the respondent lived alone or with others in the household.

Table 4 and Fig 3 summarize the mediation analysis, showing how county-level social distancing and disease and financial threats predict respondents' mental distress both directly and indirectly through their influence on respondents' social contacts with people outside their household and the worries they report having about getting ill and household finances.

**Table 1. Definitions of measures used in the analysis.**

| Level | Composite Variables | Component Measures | Contents | Contents Note | Data Source |
|---|---|---|---|---|---|
| Respondent-level outcome | Mental Distress | Anxiety Depression | In the past 5 days, how often have you felt: <br> a. nervous, anxious, or on edge? <br> b. depressed? | 4-point Likert scales (None of the time/Some of the time/Most of the time/All the time); Pearson r = .67. | [1] |
| County-level predictor | Disease Threat | Cumulative infection rates | 1. Cumulative number of COVID-19 cases per 100,000 residents. | Confirmed by state and local health authorities per day. | [2] |
| | | Daily infection rates | 2. Daily new number of COVID-19 cases per 100,000 residents. | Confirmed by state and local health authorities per day. | [2] |
| | | People known to be sick | 3. How many additional people in your local community that you know personally are sick (fever, along with at least one other symptom from the above list)? Other symptoms include sore throat, cough, shortness of breath and difficulty breathing. | Two-week moving average of the responses to this question by other respondents in the focal respondent's county. "Other respondents" refers to aggregated responses to the survey from all the other survey respondents in a focal respondent's county during a two-week period after excluding the focal respondent's response. | [1] |
| | | Average worry about disease | 4. How do you feel about the possibility that you or someone in your immediate family might become seriously ill from COVID-19 (coronavirus disease)? | Two-week moving average of the responses to this question by other respondents in the focal respondent's county. | [1] |
| County-level predictor | Social Distancing | Time spent outside home | 1. Average time per day members of the SafeGraph panel in a respondent's county spent outside their home (reverse scored). | This measure correlates moderately with the Oxford University measures of state-level stringency of policy restrictions on workplaces, schools, bars, restaurants, business, and public gatherings, as well as stay-at-home orders (r = -0.43; 20). However, we did not use the policy restriction data directly because it was not granular enough to capture week-by-week changes in social distancing. It was only available at the state and not county level, and policy restrictions did not change frequently. In contrast our measure of time spent outside the home has a finer geographic and temporal granularity. | [3] |
| | | Average social contact | 2. Two-week moving average of the "Social Contact Outside Home" (respondent-level mediator) by other respondents in the focal respondent's county (**reverse scored**). | | [1] |
| County-level predictor | Financial Threat | Unemploy-ment rates | 1. County-level Unemployment rates | Unemployment rates were updated monthly. | [4] |
| | | Average worry about finances | 2. How much of a threat would you say the coronavirus outbreak is to your household's finances? | 4-point Likert scale (A substantial threat/A moderate threat/Not much of a threat/Not a threat at all). | [1] |
| | | | | Two-week moving average of the responses to this question by other respondents in the focal respondent's county. | |
| Respondent-level mediator | Disease Worry | Disease worry | How do you feel about the possibility that you or someone in your immediate family might become seriously ill from COVID-19 (coronavirus disease)? | 4-point Likert scale (Very worried/Somewhat worried/Not too worried/Not worried at all) | [1] |

(*Continued*)

 

**Table 1.** (Continued)

| Level | Composite Variables | Component Measures | Contents | Contents Note | Data Source |
|---|---|---|---|---|---|
| Respondent-level mediator | Social Contact Outside Home | Number of direct contacts | 1. In the past 24 hours, with how many people have you had direct contact* outside of your household? Your best estimate is fine. | Number of direct contacts: At work/Shopping for groceries and other essentials/At social gatherings/Other. | 1 |
| | | | | "Direct contact" means a conversation lasting more than 5 minutes with a person who is closer than 6 feet away from you, or physical contact like hand-shaking, hugging, or kissing | |
| | | Not avoid contact | 2. To what extent are you intentionally avoiding contact with other people? (reverse scored). | 4-point Likert scale (All of the time/Most of the time; I only leave my home to buy food and other essentials/Some of the time; I have reduced the amount of times I am in public spaces, social gatherings, or at work/None of the time). | 1 |
| | | Work outside | 3. In the past 5 days, have you gone to work outside of your home? | Yes/No. | 1 |
| Respondent-level mediator | Financial Worry | Financial worry | How much of a threat would you say the coronavirus outbreak is to your household's finances? | 4-point Likert scale (A substantial threat/A moderate threat/Not much of a threat/Not a threat at all). | 1 |
| Covariate | Age | Age | What is your age? | 7 buckets (18-24/25-34/35-44/45-54/55-64/65-74/75 plus). | 1 |
| Covariate | Female | Female | What is your gender? | 5 buckets (Male/Female/Non-binary/ Prefer to self-describe/Prefer not to answer). | 1 |
| | | | | Since "Male" and "Female" represented nearly 98% of responses, we treated the other three responses as missing. | |
| Covariate | Live with someone | Live with someone | How many children under 18 years old currently stay in your household? | The number of children, adults 18–64 and adults 65+ was added together, then transformed into a binary variable, with 0 representing the respondent lives alone and 1 representing the respondent lives with at least one person. | 1 |
| | | | How many adults between 18 and 64 years old currently stay in your household (not including yourself)? | | |
| | | | How many adults 65 years old or older currently stay in your household (not including yourself)? | | |

Data sources:

[1] COVID-19 Trends and Impact Survey [17]

[2] Delphi's COVIDcast Epidata API: JHU Cases and Deaths [17]

[3] Delphi's COVIDcast Epidata API: SafeGraph [19]

[4] US Bureau of Labor Statistics: http://www.bls.gov/ [18]

Table 4 also shows the direct effects of respondents' demographics (age bracket, gender, and household composition) on their mental distress. The unit of analysis is a survey response. Because continuous variables have been standardized at the county level, the coefficients reflect the effects on respondents' mental distress in standard deviation units resulting from a change of a binary predictor from zero to one (e.g., living alone to living with others) or the increase of a continuous variable by one standard deviation from its county-level base rate. Because of the very large sample size, all coefficients are reliably different from zero at the $p < .0001$ level. The structural equation mediation model is a good fit to the data (SRMR = .006), where an SRMR less than 0.08 indicates a good model fit [22]. Multicollinearity is not a problem, as all VIF values are lower than 2.5 (mean VIF = 1.1).

Fig 3 shows both direct effects of county-level stressors on respondents' mental distress and effects mediated by respondent-level worries about disease, social contact, and household

**Table 2. Descriptive statistics.**

| Level | Measure | Components | N | Median | Mean | SD | Cronbach alpha |
|---|---|---|---|---|---|---|---|
| County | Disease threat | Cumulative infection rates | 13,974,331 | 662.097 | 1121.416 | 1265.332 | 0.60 |
| | | Daily infection rates | 13,974,331 | 7.002 | 14.663 | 26.606 | |
| | | Average worry about disease | 12,612,918 | 2.829 | 2.824 | 0.191 | |
| | | People known to be sick | 12,485,381 | 0.359 | 0.452 | 0.314 | |
| County | Social distancing | Time spent outside home | 13,974,331 | 1.551 | 1.609 | 0.526 | 0.59 |
| | | Average social contact | 13,239,223 | -0.011 | -0.014 | 0.202 | |
| County | Financial threat | Unemployment rates | 13,973,490 | 9.4 | 10.165 | 4.442 | 0.71 |
| | | Average worry about finances | 12,297,292 | 2.644 | 2.614 | 0.213 | |
| Respondent | Mental Distress | Anxiety | 12,775,238 | 2 | 1.828 | 0.868 | 0.80 |
| | | Depression | 12,582,827 | 1 | 1.663 | 0.825 | |
| Respondent | Disease worry | Disease worry | 12,612,918 | 3 | 2.824 | 0.966 | |
| Respondent | Social Contact outside home | Work outside | 13,214,089 | 0 | 0.420 | 0.494 | 0.69 |
| | | Not avoid contact | 10,364,443 | 2 | 2.389 | 0.826 | |
| | | Num of direct contacts | 11,334,164 | 3 | 15.413 | 37.754 | |
| | Financial worry | Financial worry | 12,297,292 | 3 | 2.614 | 1.019 | |
| | Age | Age | 12,436,570 | 3 | 2.892 | 1.636 | |
| | Female | Female | 12,236,326 | 1 | 0.670 | 0.47 | |
| | Live with someone | Live with someone | 12,609,199 | 1 | 0.907 | 0.291 | |

All variables reflect raw data.

finances. The three coefficients on the dotted lines respectively represent the total effect, the direct effect, and the indirect effect of county-level variables on mental distress. Note, all coefficients in Fig 3 are highly significantly different from zero, with all p-values < .0001.

## Effects of covariates

Women, younger respondents, and respondents living alone reported greater mental distress compared to men, older respondents and those living with others respectively. The associations of mental distress with these demographic characteristics are all consistent with prior

**Table 3. Correlation among variables used in the analysis (*p<0.05).**

| | Respondent-level Outcome | County-level Predictors | | | Respondent-level Mediators | | | Respondent-level Covariates | |
|---|---|---|---|---|---|---|---|---|---|
| | Mental distress | Disease threat | Social distancing | Financial threat | Disease worry | Social contact outside home | Financial worry | Age | Female |
| Disease threat | .025* | _ | | | | | | | |
| Social distancing | .017* | .357* | _ | | | | | | |
| Financial threat | .016* | .185* | .184* | _ | | | | | |
| Disease worry | .289* | .051* | .031* | .024* | _ | | | | |
| Social contact Outside home | -.052* | -.042* | -.067* | -.021* | -.231* | _ | | | |
| Financial worry | .244* | .021* | .012* | .053* | .25* | .008* | _ | | |
| Age | -.245* | -.006* | .003* | -.009* | -.045* | -.226* | -.130* | _ | |
| Female | .149* | .004* | .006* | .005* | .161* | -.138* | .062* | -.047* | _ |
| Live with someone | .028* | .004* | -.001* | -.001* | -.009* | .080* | .058* | -.188* | .006* |

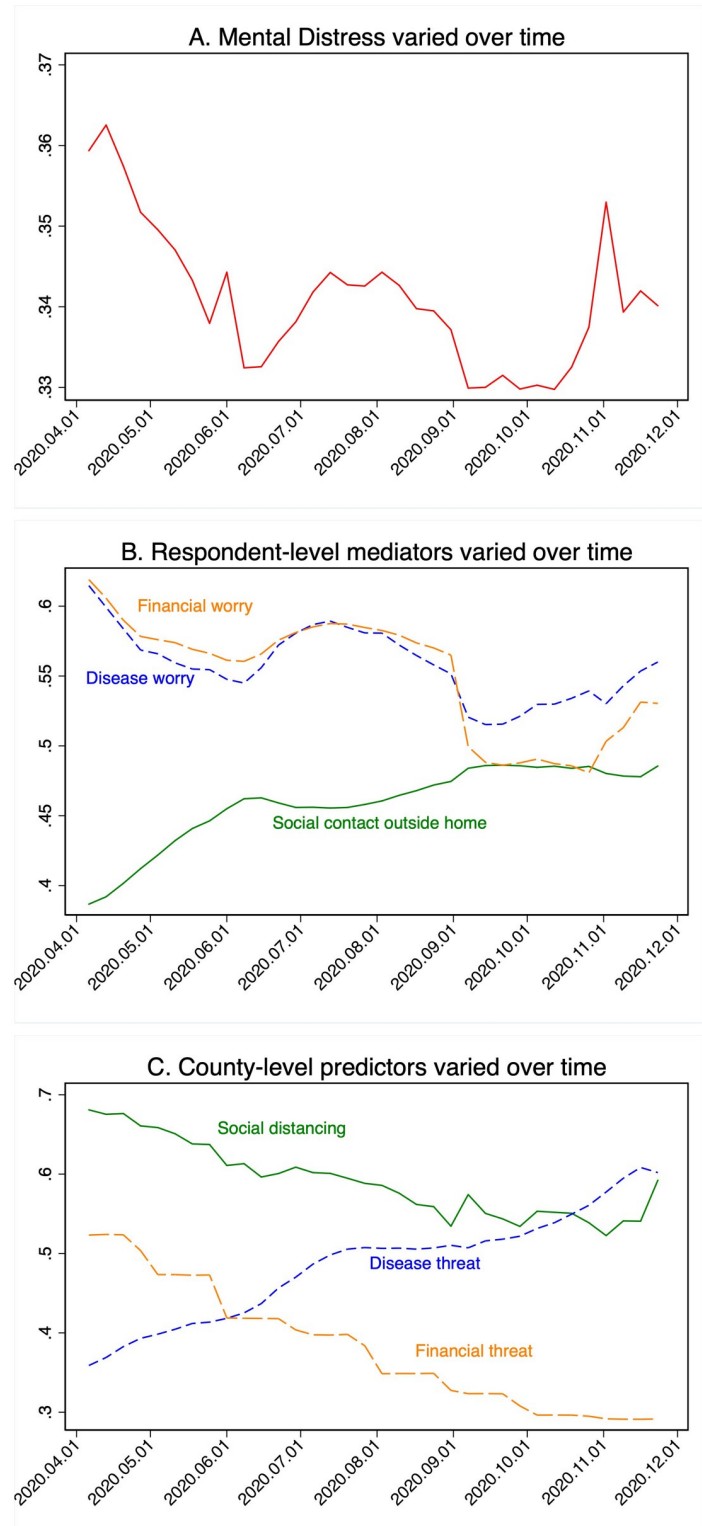

**Fig 2. Mean level of the variables of interest over time.** (A) Self-reported mental distress. (B) Respondent-level endogenous mediating variables (disease and financial worries and social contact). (C) County-level exogenous predictor variables (disease and financially-related stressors and social distancing). All variables were normalized to the range of 0 to 1 before averaging. Averages vary over both location and time.

**Table 4. Mediation analysis predicting mental distress from county-level and respondent-level variables.**

| | | Mental distress | | | | | | | | | | | | | | |
|---|---|---|---|---|---|---|---|---|---|---|---|---|---|---|---|---|
| | | Total Effect | | | | | Direct Effect | | | | | Indirect Effect | | | | |
| Level | Predictor | β | S.E. | t | p | 95% C.I. | β | S.E. | t | p | 95% C.I. | β | S.E. | t | p | 95% C.I. |
| County | Disease threat | .013 | .0005 | 29.82 | 0.000 | .013-.014 | .005 | .0004 | 11.29 | 0.000 | .004-.006 | .0085 | .0001 | 87.94 | 0.000 | .008-.009 |
| County | Social distancing | .011 | .0005 | 24.31 | 0.000 | .01-.012 | .004 | 0.0004 | 9.95 | 0.000 | .004-.005 | .0068 | .0001 | 48.13 | 0.000 | .006-.007 |
| County | Financial threat | .004 | .0004 | 9.61 | 0.000 | .003-.005 | -.005 | 0.0004 | -10.72 | 0.000 | -.005--.004 | .0088 | .0001 | 117.87 | 0.000 | .0086-.0089 |
| Respondent | Disease worry | .205 | .0005 | 434.31 | 0.000 | .204-.206 | .205 | 0.0005 | 434.31 | 0.000 | .204-.206 | NA | NA | NA | NA | NA |
| Respondent | Social contact | -.042 | .0005 | -91.68 | 0.000 | -.043--.041 | -.042 | 0.0005 | -91.68 | 0.000 | -.043--.041 | NA | NA | NA | NA | NA |
| Respondent | Financial worry | .158 | .0005 | 349.67 | 0.000 | .157-.159 | .158 | 0.0005 | 349.67 | 0.000 | .157-.159 | NA | NA | NA | NA | NA |
| Respondent | Age | -.149 | .0003 | -512.93 | 0.000 | -.149--.148 | -.138 | 0.0003 | -480.49 | 0.000 | -.139--.138 | -.0104 | .0001 | -92.39 | 0.000 | -.011- -.01 |
| Respondent | Female | .314 | .0009 | 361.59 | 0.000 | .313-.316 | .208 | 0.0009 | 240.65 | 0.000 | .206-.21 | .1063 | .0003 | 324.86 | 0.000 | .106-.107 |
| Respondent | Live with someone | -.048 | .0009 | -51.45 | 0.000 | -.05- -.047 | -.051 | 0.0009 | -57.32 | 0.000 | -.053--.049 | .0028 | .0003 | 9.24 | 0.000 | .002-.003 |

The significance of the mediation effects was calculated using the Sobel test [23].

research. For example, Salk et al's meta-analysis revealed that women are substantially more likely to be diagnosed with depression and show depressive symptoms across multiple cultures [24]. Abundant research also demonstrates the links between social integration and both physical and mental health [25]. For example, people who live alone are almost twice as likely to report symptoms of depression compared to those who live with others in their household [26]. Although not the focus of the current project, survey findings showing less mental distress among respondents in households with other adults or children are consistent with one of our core hypotheses: social contact is a preventative for mental distress. However, because

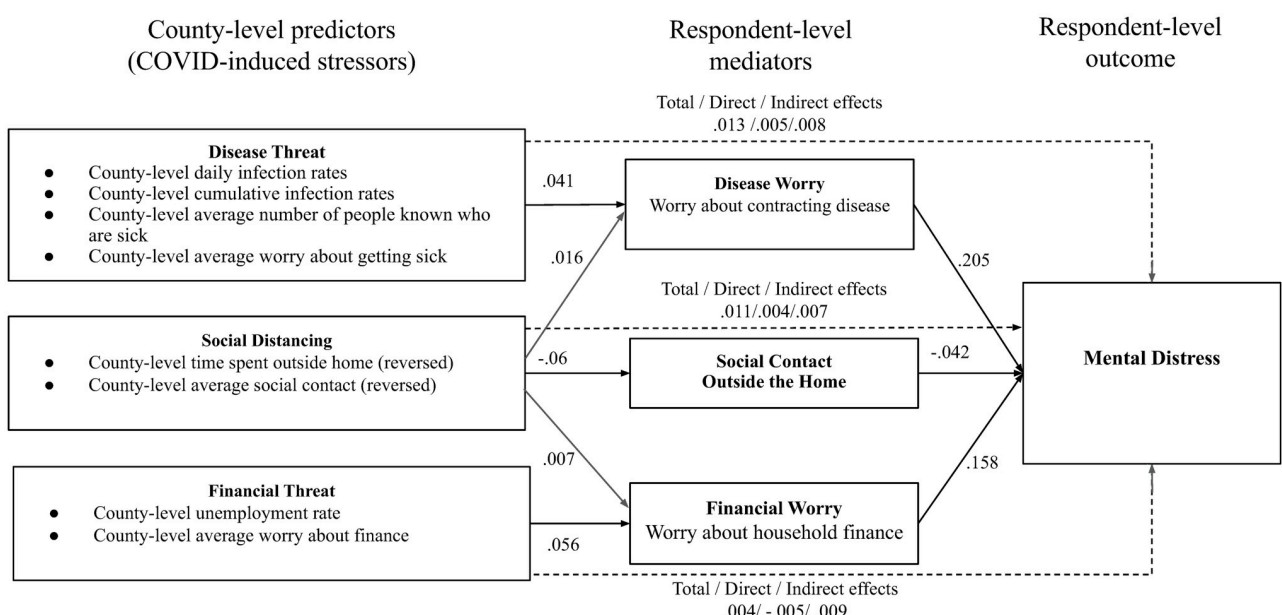

**Fig 3. Path diagram showing the direct and indirect effects of county-level stressors on respondents' mental distress.**

household composition is something that people choose, and people who are predisposed to mental distress may be more likely to choose to live alone, the association of household composition and mental distress is not itself strong evidence of a causal relationship between social contact and mental health. In contrast, an association of social distancing at the county level with mental distress is stronger causal evidence of the role of social contact, because it is unlikely that respondents with greater than average mental distress choose to live in a county during a period in which social distancing is greater than average.

## Effects of disease threat, social distancing, and financial threat on respondents' mental health

**Disease threat.** The mediation analysis in Table 4 shows that overall greater threat of disease in an area is associated with more mental distress ($\beta$ = .013, p = 0.000). Because the county-level disease threat is associated with individual respondents' worries that they or a family member would get ill from the disease ($\beta$ = .041, p = 0.000), and these worries about the disease are associated with their mental distress ($\beta$ = .205, p = 0.000), nearly 65% of the total effect of county-level disease threat is mediated by individual worries about disease ((.041 x .205)/.013). This mediation result suggests that objective risks of illness have their effects on mental distress to a large extent by elevating individual worries about becoming ill.

**Social distancing.** Overall, social distancing at the county level is associated with increased mental distress ($\beta$ = .011, p = 0.000). Because county-level social distancing is associated with individual respondents' having less social contact outside the home ($\beta$ = -.06, p = 0.000), more disease worries ($\beta$ = .016, p = 0.000), and more financial worries ($\beta$ = .007, p = 0.000) and because these individual level behaviors and worries are associated with mental distress ($\beta$ = -.042, .205, and .158 respectively, p = 0.000), the mediation analysis suggests that social distancing in a county affects respondents' mental distress to a large extent by reducing their contact with others outside their homes, as well as increasing their worries about disease and finances. Approximately 63% of the total effect of county-level social mobility on mental distress is indirect and mediated by respondents' self-reported reductions in social contact outside of their households, and disease and financial worries (((-.06 x -.042) + (.016x.205) + (.007x.158))/.011 = .628).

The total effect of county-level social distancing on mental distress is small but practically important because it is a policy lever at the disposal of government authorities, and because of the many millions of people whose mental health might be affected by social distancing policies and norms. The small effect size may be the result of other factors that influence respondents' social mobility besides objective social distancing policies and norms, such as whether they are employed outside the home, their gender, or their household composition, which can also influence their mental distress.

In addition, lock-down orders and other reductions in opportunities for social contact outside the household may cause people to spend more time interacting with others in their household and thus compensate for the impact of social distancing on mental health. That is, for people who live with others, the greater social contact they have within the home may compensate for reductions in social contacts outside it. To examine this possibility, we tested whether the mental health benefits of greater county-level social mobility were greater for respondents living alone than for those living with another in their household. We added an interaction term between *social distancing* and *live with someone* to the SEM model. S2 Table shows the results. As a reminder, 76% of respondents lived with at least another adult or a child and 24% lived alone. The lack of a statistical interaction between county-level social distancing and living with others suggests that social distancing was *not* more harmful in terms of

increasing mental distress among people who live alone ($\beta$ = -.002, p = 0.089). The marginal analysis shows that a standard deviation increase in county-level social distancing was associated with a .0044 standard deviation increase in mental distress both among those who lived alone and those living with another person.

**Financial threat.**   Overall, financial threat at the county level was associated with more mental distress ($\beta$ = .004, p = 0.000). This result is consistent with Witteveen and Velthorst's findings in six European countries showing a "positive relationship between instantaneous economic hardships during the COVID-19 lockdown and expressing feelings of depression and health anxiety" [16]. A systematic review of research on COVID-19-related fear and anxiety and job-related outcomes also shows that fear of COVID-19 was associated with increased future career anxiety, decreased job satisfaction, and perceived job insecurity [27]. Because county-level financial threats were associated with respondents worrying more about household finances ($\beta$ = .056, p = 0.000), and respondent-level financial worries are in turn associated with greater mental distress ($\beta$ = .158, p = 0.000), the mediation analysis suggests that county-level financial threat affects respondents' mental distress partially by increasing their personal worry about household finances ($\beta$ = .009, p = 0.000). Surprisingly, though, the direct effect of county-level financial threats was to reduce mental distress ($\beta$ = -.005, p = 0.000). To rule out the possibilities of multicollinearity due to a correlation between social distancing and county-level financial threats ($r$ = .184), we conducted an additional analysis excluding social mobility. The result shows a robust negative effect of county-level financial threats on mental distress ($\beta$ = -.004, p = 0.000). If county-level financial threats raised personal financial concerns which in turn increased mental distress, why was the direct effect of county-level financial threats to reduce mental distress? It may be that even though unemployment and concerns about finances in the county exacerbated mental distress by raising personal worries about finances, the 2020 Coronavirus Aid, Relief, and Economic Security Act (CARES Act) and other government stimulus programs reduced the actual financial pain associated with the pandemic, but these effects of government stimulus programs were not reflected in the county-level employment data we had available. Moreover, considering the overrepresentation of adults with higher education in the survey samples, those highly educated individuals might have been in white collar occupations less subject to the economic hardships caused by the pandemic.

## Discussion

This research provides evidence consistent with the thesis that the COVID-19 pandemic harmed the mental well-being of adults in the United States and identifies specific stressors associated with the pandemic that seem responsible for increasing mental distress. The current study distinguished objective stressors from respondents' perceptions of stress, thereby reducing common method biases that have inflated the associations between stressors and mental distress in earlier research. Note that mental distress correlates ten times more highly with these measures of disease and finances worries and social contact, all of which were measured by self-report at the respondent level (mean absolute correlation = .195), than with parallel measures of disease and financial threat and social distancing measured at the county level (mean absolute correlation = .019). This substantial difference in strength of association is consistent with the speculation that Kämpfen et al's results [9] were inflated by common method variance.

The analyses show that objective disease stressors outside of respondents' control, as measured by county-level COVID-19 infection rates and the likelihood of other county residents knowing someone with COVID symptoms and being worried about getting ill, increased

individuals' mental distress in part by increasing their own worries that they or other family members would get ill. Overall objective financial stressors, as measured by county-level unemployment rates and county-level perceptions that the pandemic was harming household finances, were associated with *greater* mental distress, and these effects were mediated by the extent to which these financial stressors caused respondents to become worried about harm to their own household's finances. However, the direct effect of the financial stressors seemed to be to *lower* mental distress. Future research is needed to examine the mechanism for this direct effect; for example, did federal government stimulus programs reduce the actual financial pain and associated mental distress during times of high unemployment, and what role did sociodemographic variables play?

The reduction in social contact caused by the pandemic is especially interesting. Reductions in social contact outside of the household, which were partially caused by official lock-down orders and informal norms in a county, were associated with greater mental distress. Authorities issued shutdowns and stay-at-home orders and people voluntarily reduced outside social contact to reduce their risk of becoming infected with the disease, and to the extent that these actions reduced people's worries about getting COVID, they also reduced mental distress. However, to the extent that these efforts also reduced people's social contact outside the home, they had the undesirable side effect of increasing mental distress. Surprisingly, the harmful effects of social distancing policies and behavioral norms in increasing mental distress applied equally to those who lived alone and those who lived with others. This latter result is consistent with previous research showing benefits from social interaction with coworkers, acquaintances, and other weak ties even among people who have greater than average strong-tie interactions [28] and benefits from social interactions in social spaces outside the home [29]. We believe one can balance the benefits of reduced social contact to slow the spread of disease with the mental health harms to social isolation. For example, having social interactions online, through text chats, emails, phone calls, and video sessions may successfully substitute for in-person social interactions. Very little research has examined the impact that modality of communication has on mental well-being [30].

This study has several methodological limitations that should be considered when interpreting the findings. Respondents' mental distress was measured by two questions assessing depression and anxiety. Follow-up research should use more robust and clinically validated measures of psychological distress. While the survey data were representative of US adults in terms of age, gender and region, there was a sampling bias associated with education; the survey respondents were more educated than average US citizens. Additionally, follow-up research should examine how community-level predictors, like poverty and social inequality, and individual-level ones, like occupational status, moderate the stresses associated with the pandemic.

Despite these limitations, this research allows us to make stronger causal claims than possible with more conventional, respondent-level survey research about the impact of COVID-related stressors on mental health, because it examines pathways through which pandemic-related reductions in social mobility and increases in disease and financial stressors, measured at the county level and therefore out of respondents' control, influence respondent-level social contacts and worries, which in turn lead to increases in mental distress.

Although the focus of this research was to understand how pandemic-related stressors were influencing mental health during the COVID-19 pandemic, it also extends our theoretical understanding of how social support works. Decades of research have provided strong and consistent evidence that social ties and social support improve many aspects of personal health, including all-cause mortality [31, 32], physical health [33, 34] and mental health [35], but the mechanisms are still murky [25]. It is unclear whether the component of support that is most

valuable is the perception that support will be available when needed (i.e., perceived support), the explicit exchange of support during times of stress, especially from strong ties, or merely the accumulation of everyday social interactions [25, 36–38]. Results of the current research are consistent with the thesis that mundane social interactions can lead to well-being. Week-to-week changes in the frequency of social contacts in the community seem to lead to changes in respondents' mental distress, suggesting that to some extent it is social interactions that people actually engage in outside the home that confer benefits, rather than slower-to-change perceived social support. While our data show that people who are living with others have less mental distress, they also show that social contact outside of the household, and presumably with less intimate ties, also confers benefits over and above social contact within the home.

## Data sharing

Most of the data reported in this paper are publicly available in the Epidata API, maintained by the Delphi research group at Carnegie Mellon University and available through R and Python clients. These data come from multiple sources, with data licensing handled separately for each source. The de-identified survey data are available to researchers associated with universities or non-profit organizations. Researchers who want access to the survey data should submit an information request on Facebook's COVID-19 Symptom Survey–Request for Data Access page [39].

## Supporting information

**S1 Table. Characteristics of the study sample, compared to 2019 American community survey supplemental estimates.**
(PDF)

**S2 Table. Mediation analysis predicting mental distress from county-level and respondent-level variables, including the interaction between *Social distancing* and *Live with someone*.**
(PDF)

## Acknowledgments

The authors would like to thank Alex Reinhart for statistical advice and Aya Betensky for editorial assistance.

## Author Contributions

**Conceptualization:** Robert E. Kraut, Han Li, Haiyi Zhu.

**Formal analysis:** Han Li.

**Methodology:** Robert E. Kraut.

**Supervision:** Robert E. Kraut, Haiyi Zhu.

**Writing – original draft:** Robert E. Kraut, Han Li, Haiyi Zhu.

**Writing – review & editing:** Robert E. Kraut, Haiyi Zhu.

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
