## [Decision Letter · Decision Letter 0]

16 May 2022

PONE-D-22-07712Mental health during the COVID-19 pandemic: Impacts of disease, social isolation and financial stressorsPLOS ONE

Dear Dr. Kraut,

Thank you for submitting your manuscript to PLOS ONE. After careful consideration, we feel that it has merit but does not fully meet PLOS ONE’s publication criteria as it currently stands. Therefore, we invite you to submit a revised version of the manuscript that addresses the points raised during the review process.

We look forward to receiving your revised manuscript.

Kind regards,

Mohammad Farris Iman Leong Bin Abdullah, Dr Psych

Academic Editor

PLOS ONE

Journal Requirements:

Additional Editor Comments (if provided):

1. Please ensure that your manuscript meets PLOS ONE's style requirements, including those for file naming. The PLOS ONE style templates can be found at:

Reviewers' comments:

Reviewer's Responses to Questions

**Comments to the Author**

1. Is the manuscript technically sound, and do the data support the conclusions?

Reviewer #1: Partly

Reviewer #2: Yes

2. Has the statistical analysis been performed appropriately and rigorously? 

Reviewer #1: No

Reviewer #2: I Don't Know

3. Have the authors made all data underlying the findings in their manuscript fully available?

Reviewer #1: Yes

Reviewer #2: Yes

4. Is the manuscript presented in an intelligible fashion and written in standard English?

Reviewer #1: Yes

Reviewer #2: Yes

5. Review Comments to the Author

Reviewer #1: The study is well written and is important during the COVID-19 pandemic. However, there are some issues that should be addressed in a revised manuscript. Please see my comments below.

1. Introduction should be strengthened using some recent systematic reviews on the psychological distress during COVID-19 pandemic.

Rajabimajd N, Alimoradi Z, Griffiths MD. Impact of COVID-19-related fear and anxiety on job attributes: A systematic review. Asian J Soc Health Behav 2021;4:51-5

Olashore AA, Akanni OO, Fela-Thomas AL, Khutsafalo K. The psychological impact of COVID-19 on health-care workers in African Countries: A systematic review. Asian J Soc Health Behav 2021;4:85-97

Alimoradi, Z., Ohayon, M. M., Griffiths, M. D., Lin, C.-Y., & Pakpour, A. H. (2022). Fear of COVID-19 and its association with mental health related factors: A systematic review and meta-analysis. BJPsych Open, 8, e73.

Alimoradi, Z., Gozal, D., Tsang, H. W. H., Lin, C.-Y., Broström, A., Ohayon, M. M., & Pakpour, A. H. (2022). Gender-specific estimates of sleep problems during the COVID-19 pandemic: Systematic review and meta-analysis. Journal of Sleep Research, 31(1), e13432.

Alimoradi, Z., Broström, A., Tsang, H. W. H., Griffiths, M. D., Haghayegh, S., Ohayon, M. M., Lin, C.-Y., Pakpour, A. H. (2021). Sleep problems during COVID-19 pandemic and its’ association to psychological distress: A systematic review and meta-analysis. EClinicalMedicine, 36, 100916.

2. It seems to me that the authors have collected data with multilevel (i.e., country level and respondent level); however, the SEM described in the Data analysis section does not mention how the multilevel was treated in the multilevel data. Moreover, the authors mentioned that they examined the mediation effects. However, there is no information regarding how they determined whether the mediation effects are significant or not. Did the authors use Sobel test or bootstrapping? Accordingly, quite a lot of information is not described for the SEM, such as the fit indices, the estimator used in the model, and how the authors treated the variables as latent or observed ones.

3. The authors have done more than SEM in the data analysis; however, they only described their data analysis in SEM.

Reviewer #2: The mental health effects of the ongoing Covid 19 pandemic is a matter of concern. The present study is significant in its aim to measure the impact of various stressors, and its findings can help in addressing mental health challenges in future pandemics. However, there are a few concerns I hope the authors would address-

1. The manuscript is too lengthy and confusing at times. I would suggest a stringent editing with more clarity in the paper.

2. At many places in the manuscript there is mention of "other respondents". Does it refer to the respondents of the survey? If not, please clarify.

3. In the paragraph County level predictors : Disease threat : People known to be sick: the sentence ''

This was measured at the county level as the two-week moving average of the number of people other respondents reported on the survey whom they knew had COVID-19 symptoms.'' is confusing. Were they Covid 19 patients or is it people who might be having Covid 19 or similar symptoms but not Covid 19.

4. In the paragraph ''Effects of disease threat, social distancing, and financial threat on respondents’ mental

health.

Disease threat."

The authors need to clarify if the mental stress due to disease threat also corelated to already existing high rate of infection in the county or was it just related to generalised worry among individual respondents about the possibility of falling ill.

6. PLOS authors have the option to publish the peer review history of their article (what does this mean?). If published, this will include your full peer review and any attached files.

Reviewer #1: No

Reviewer #2: No

---

## [Author Response · Author response to Decision Letter 0]

8 Jul 2022

We have included detailed responses to reviewers in the file 'PLOS-ResponseLetter-v5-ab.docx.' We have pasted in the text from the response letter below. 

Response to Reviewers

Reviewer 1: 

Introduction should be strengthened using some recent systematic reviews on the psychological distress during COVID-19 pandemic.

We appreciate these pointers to the newer reviews and have incorporated all of them at appropriate places in the introduction. 

It seems to me that the authors have collected data with multilevel (i.e., country level and respondent level); however, the SEM described in the Data analysis section does not mention how the multilevel was treated in the multilevel data.

Reviewer1 is correct that we have collected multi-level data, with respondent nested within county, but our original submission did not use multi-level structural equation modeling (SEM) or regression techniques to analyze the data. In re-examining our data in response to this critique, we tried to conduct multi-level SEM models in both Stata, using the gsem command, and in R, using the lavaan package. However, these multi-level SEM models failed to converge, even after allowing them to run for multiple days, because of the large size of our dataset, with over 11.6 million observations, the large number of clusters, with 3,214 counties, and the highly uneven distribution of observations across counties, with the 10% smallest counties contributing fewer than 134 observations over the 239 observation periods while the 10% largest counties contributed 928 or more per county. 

We chose to use SEM because it is the modern way to conduct mediation analyses and it provides a compact, reader-friendly representation of the results, which simultaneously takes into account the multiple relationship between exogenous county-level predictors, individual-level mediators, and the outcome, showing both direct and indirect effects. However, as a robustness check to see whether the single-level SEM model was producing misleading results, we partially replicated the analysis using a series of multi-level one-mediator-at-a-time regressions analyses, using the R lmer package. For example, instead of computing all the relationship between exogenous county-level predictors, individual-level mediators, and individual-level distress simultaneously, the simplified analysis examined the direct and indirect effect of exogenous country-level social distancing on distress as mediated by individual-level worries about the disease (equation 3) and about finances (equation 5) separately. This robustness check shows that the single-level SEM mediation analysis presented in our paper has the same pattern of results as the set of simplified multi-level mediation analyses. In particular, all the significant direct effects, mediation effects, and total effects revealed in the single-level SEM models multi-level also appeared in the set of simplified multi-level mediation analyses (see Table R1 below, which compares the two analyses.)

Table R1: Comparison of single-level multi-equation SEM mediation model with multi-level single mediator at a time models is included in the response letter

In addition, at the conceptual level, we think the non-independence of respondents from the same county is not a problem for three reasons. 

First, centering all variables at the county-level by subtracting out the county mean removed all static differences among counties that could affect the respondents from the county in a similar way (e.g., overall employment rate, population density, or ethnic composition). All that remains are county-level differences that vary from day-to-day (e.g., changes in unemployment, infection rates, or social distancing). 

Second, unlike the classic case of multi-level modeling, in which students are clustered in the same classroom, the sample of respondents from the same county are unlikely to communicate with and directly influence each other. These reasons for discounting most sources of non-independence are confirmed by a supplementary Intraclass Correlation (ICC) analysis, which shows that the ICC for distress is essentially zero (ICC=.1.709e-09x) and a model with a random intercept for county is not a better fit to the data than one without the random intercept (AIC with random intercept = 33982969 vs AIC without a random intercept = 33982967, log likelihood ratio = .0204, p = .8864).

In addition, recent research suggests that single-level and multi-level analyses tend to lead to the same conclusions with mental-health outcomes, even when the grouping factor is the provider (Warmerdam et al., 2019). Since multi-level modeling primarily effects the standard error of a coefficient and not its absolute value (Clarke, 2008), the standard errors and significance levels in our analyses will not be affected in a meaningful way because of the millions of observations in our sample.

Reviewer 1 also asked about testing significance level for the mediation models. “The authors mentioned that they examined the mediation effects. However, there is no information regarding how they determined whether the mediation effects are significant or not. Did the authors use Sobel test or bootstrapping? Accordingly, quite a lot of information is not described for the SEM, such as the fit indices, the estimator used in the model, and how the authors treated the variables as latent or observed ones.”

Significance levels were computing using the Sobel test, and goodness of fit using the Standardized Root Mean Square Residual (SRMR) metric. According to (Hu & Bentler, 1999) a value less than .08 is considered a good fit. In the current study, the SRMR value is .006. They are now reported in the paper.

3. The authors have done more than SEM in the data analysis; however, they only described their data analysis in SEM.

In revision we provided more detail about how we tested and rejected the hypothesis that social isolation would have a larger effect on mental distress for people who live alone by examining the statistical interaction. (See page 23.) Follow-up analyses show that a standard deviation increase in county-level social distancing was associated with a .0044 standard deviation increase in mental distress both among those who lived alone and those living with another person. We include the full interaction analysis as Table 2 in the supplementary materials. 

Reviewer 2:

1. The manuscript is too lengthy and confusing at times. I would suggest a stringent editing with more clarity in the paper.

We did another pass on editing the paper and had colleagues read it. We created a new Table 1 in the manuscript to define our major variables. This makes the manuscript shorter and clearer by consolidating the large amount of text from Table 1 in the original manuscript and the description of survey items originally in a table in the supplementary materials. However, we did not do a major overhaul of the paper because we could not identify other major areas that were too long or unclear. If this remains a point of concern, it would be very helpful if R2 could provide more guidance on which portions of the paper he/she thought were too long or lacking clarity. 

2. At many places in the manuscript there is mention of "other respondents". Does it refer to the respondents of the survey? If not, please clarify.

At first mention of “other respondents,” we described the algorithm we used to compute this value. "Other respondents” refers to aggregated responses to the survey from all the other survey respondents in a focal respondent’s county during a two-week period after excluding the focal respondent. We revised the descriptions of the variables to make it clearer. See Table 1 on page 11.

3. Disease threat: People known to be sick. The sentence ''This was measured at the county level as the two-week moving average of the number of people other respondents reported on the survey whom they knew had COVID-19 symptoms.'' is confusing. Were they Covid 19 patients or is it people who might be having Covid 19 or similar symptoms but not Covid 19?

Because one cannot expect lay survey respondents to accurately diagnose Covid, the survey listed a set of five symptoms associated with Covid (high fever, sore throat, cough, shortness of breath, difficulty breathing) and asked how many people in the respondent’s household and in the larger community were sick with a fever and at least one other symptom from this list. In revision, we revised the description of the variable People known to be sick. See Table 1 on page 11.

4. In the paragraph ''Effects of disease threat, social distancing, and financial threat on respondents’ mental health. Disease threat," the authors need to clarify if the mental stress due to disease threat also correlated to already existing high rate of infection in the county or was it just related to generalised worry among individual respondents about the possibility of falling ill?

 As described in the paper, total greater disease threat in a county was associated with more mental distress among the individual respondents (β=.013). Nearly 65% of the total effect of county-level disease threat was mediated by individual worries about disease. 

Note that county-level disease threat was measured at the county level by using a composite index that included the county-level cumulative COVID infection rate and daily rate of new COVID cases per 100,000 in the population, as well as the measures from other respondents’ concerns about becoming seriously ill from COVID and the number of people they knew who were sick in the focal respondent’s county. The Cronbach alpha of .60 shows that these four measures were moderately correlated with each other. We do not report in the paper how the county-level cumulative COVID infection rate alone associates with mental distress.

---

## [Decision Letter · Decision Letter 1]

31 Oct 2022

Mental health during the COVID-19 pandemic: Impacts of disease, social isolation and financial stressors

PONE-D-22-07712R1

Dear Dr. Robert E. Kraut,

We’re pleased to inform you that your manuscript has been judged scientifically suitable for publication and will be formally accepted for publication once it meets all outstanding technical requirements.

Kind regards,

Mohammad Hayatun Nabi, MBBS, MHSM, MPH, PHD

Academic Editor

PLOS ONE

Additional Editor Comments (optional):

Reviewers' comments:

Reviewer's Responses to Questions

**Comments to the Author**

1. If the authors have adequately addressed your comments raised in a previous round of review and you feel that this manuscript is now acceptable for publication, you may indicate that here to bypass the “Comments to the Author” section, enter your conflict of interest statement in the “Confidential to Editor” section, and submit your "Accept" recommendation.

Reviewer #1: All comments have been addressed

Reviewer #2: All comments have been addressed

2. Is the manuscript technically sound, and do the data support the conclusions?

Reviewer #1: Yes

Reviewer #2: Yes

3. Has the statistical analysis been performed appropriately and rigorously? 

Reviewer #1: Yes

Reviewer #2: I Don't Know

4. Have the authors made all data underlying the findings in their manuscript fully available?

Reviewer #1: Yes

Reviewer #2: Yes

5. Is the manuscript presented in an intelligible fashion and written in standard English?

Reviewer #1: Yes

Reviewer #2: Yes

6. Review Comments to the Author

Reviewer #1: The authors have addressed all my previous concerns and I am happy with the present version of submission. I have no more comments for the revised manuscript.

Reviewer #2: All the queries raised have been addressed adequately. The inclusion of Table 1 has given more clarity.

7. PLOS authors have the option to publish the peer review history of their article (what does this mean?). If published, this will include your full peer review and any attached files.

Reviewer #1: No

Reviewer #2: No

---

## [Editor Report · Acceptance letter]

11 Nov 2022

PONE-D-22-07712R1 

Mental health during the COVID-19 pandemic: Impacts of disease, social isolation, and financial stressors 

Dear Dr. Kraut:

I'm pleased to inform you that your manuscript has been deemed suitable for publication in PLOS ONE. Congratulations! Your manuscript is now with our production department. 

Kind regards, 

on behalf of

Dr. Mohammad Hayatun Nabi 

Academic Editor

PLOS ONE